# Factors Influencing Sexuality in Late Adolescence: A Qualitative Study on Heterosexual Adolescents’ Perspectives

**DOI:** 10.3390/healthcare11233032

**Published:** 2023-11-24

**Authors:** Isabel María Fernández-Medina, Miguel Angel Ramal-Gea, María Dolores Ruíz-Fernández, José Granero-Molina, María del Mar Jiménez-Lasserrotte, María Isabel Ventura-Miranda

**Affiliations:** 1Department of Nursing, Physiotherapy and Medicine, University of Almería, 04120 Almería, Spain; miguelangelrage94@gmail.com (M.A.R.-G.); md.ruizfernandez@ual.es (M.D.R.-F.); jgranero@ual.es (J.G.-M.); mjl095@ual.es (M.d.M.J.-L.); mvm737@ual.es (M.I.V.-M.); 2Faculty of Health Sciences, Universidad Autónoma de Chile, Temuco 4780000, Chile

**Keywords:** sexuality, late adolescence, sexual education, experiences, qualitative study

## Abstract

(1) Background: Sexuality is influenced by the school, family, and social contexts. All of these play a crucial role in promoting adolescents’ sexual health and well-being. However, little is known about the factors that have an impact on sexuality in late adolescence. The aim of this study was to explore and describe the perspectives of late adolescents on the factors that influence their sexual lives. (2) Methods: A qualitative study with a descriptive design was carried out. Thirteen interviews were conducted. Data were audio-recorded, transcribed, and categorized into themes and subthemes. (3) Results: The results of our study offer support for the importance of sexual aspects in the lives of late adolescents who identify as heterosexual. (4) Conclusions: The most influential environments in late adolescent sexuality are the social, family, school, and peer contexts.

## 1. Introduction

Sexuality, which accompanies us throughout our life cycle, is an integral part of adolescent health, with great importance for their physical, emotional, and social development [1]. Late adolescence is a transitional period to young adulthood between the ages of 17 and 21 years characterized by a high propensity for sexual activity [2]. It is during this period that sexual activity typically begins [3], and some studies have linked sexual experience to adolescent psychosocial adjustment [4]. According to the World Health Organization (WHO), “sexuality encompasses sex, gender identities and roles, eroticism, pleasure, intimacy, reproduction and sexual orientation” [5]. However, sexuality is a particular part of each person, and it is lived and expressed in unique ways. The development of a healthy sexuality is a key point in the development of all adolescents, and it depends on the acquisition of information, attitudes, beliefs, and values about sexual orientation, gender identity, relationships, and intimacy [6]. The development of a healthy sexuality can be defined as the process through which adolescents build competencies in the form of knowledge, skills, and attitudes that support sexual well-being in relation to themselves and others [7]. This development does not occur in isolation as sexuality is influenced by the context in which the adolescent lives and, therefore, by the interaction of biological, psychological, social, cultural, religious, and ethical factors [7,8]. 

A healthy sexuality is influenced by personal, religious, cultural, and moral concerns [6]. In most cultural settings, adolescents grow up in a context characterized by multiple sexuality and gender norms and messages [7]. Perceived social norms appear to influence sexual behavioural intentions [9]. Several studies have shown how interpersonal relationships, such as those with parents, siblings, and peers, shape adolescents’ sexual behaviour and relationships [10,11,12]. Parents play a critical role in raising individuals who are able to engage in healthy intimacy and sexual behaviour [11]. Parent–adolescent sexual communication is an effective strategy for protecting adolescents from risky sexual behaviour. However, during adolescence and young adulthood, independence and autonomy from the family increase and peer relationships become more important [13]. Furthermore, sexual education is more than teaching adolescents about the physiology of biological sex and reproduction. It includes gender identity, intimacy, sexual development, body image, and interpersonal relationships [9,13,14]. Adolescents’ sexual health behaviour is strongly influenced by the behaviour of other peers, which can lead to risky sexual behaviour [9]. In addition to the role of family and peer groups, social networks are one of the most important agents of sexual socialization among adolescents. Although the internet and social networks allow adolescents to obtain information about sexuality, they can also have a negative impact on their sexual activity and normalize abnormal sexual behaviour [14]. For all of the above reasons, it is necessary to know the experiences and opinions of adolescents in order to provide guidance and research aimed at improving adolescents’ reproductive health and rights. The aim of this study was, therefore, to explore and describe the perspectives of late adolescents on the main factors influencing the development of their sexual life. 

## 2. Materials and Methods 

### 2.1. Study Design

A descriptive study was carried out to explore the experiences of adolescents. The descriptive qualitative methodology provides a better understanding of a particular phenomenon and the subjective reality of a population [15], making it suitable for this study.

### 2.2. Participants

A purposive sampling technique was used to recruit participants. Information about the study was disseminated online using social networks such as Instagram. Those adolescents who were interested in participating in the study contacted the third author via email to schedule an in-person meeting. The inclusion criteria were to be between 17 and 21 years old at the time of the interview, to speak Spanish, and to voluntarily agree to participate in the study. Participants with psychological disorders and those who refused to participate in the study were excluded. Sixteen adolescents showed interest in taking part in the study, but, finally, three of them refused to participate because of intimacy reasons. Thus, a total of 13 adolescents took part in the study: seven females and six males, all of them with a heterosexual orientation. Seven of them had an open relationship, five a closed relationship, and one did not have any kind of relationship. The characteristics of interviewed participants are shown in Table 1. 

### 2.3. Data Collection

Data collection was completed between January and February 2022 via individual semi-structured interviews. The interviews were conducted face-to-face by the third author of the study, who had experience in qualitative research, and took place in a private meeting room of the local university. We used an interview guide with open-ended questions belonging to different aspects of sexuality in adolescents based on a literature review (Appendix A). The interviews began with the collection of sociodemographic information and an introductory question about the different factors that have influenced their sexual relationships. Each participant was only interviewed once, and interviews lasted an average of 40 min. Data collection was stopped after reaching data saturation, that is, when no new information emerged. All interviews were conducted in Spanish and were digitally audio-recorded. During the interviews, a reflexive journal was used to register thoughts and observations. The reflexive journal was incorporated into the data analysis. All interviews were transcribed verbatim in Spanish and afterwards translated into English by the third author. 

### 2.4. Data Analysis

The qualitative analysis software ATLAS.ti version 9.0 was used for data analysis, organization, and coding. Researchers selected significant fragments as citations, and organized and gave them a code that described their significance. During the coding process, numerous memos were written to identify meaningful segments of the data and their relationships. The similarities and differences within the data in the interpretative phase of the analysis were established by the researchers [16]. According to the similarities, significant fragments of data were grouped into themes and subthemes by the first and last authors. During this stage, a table with units of meaning, themes, and subthemes was elaborated (Table 2). Finally, the analysis was verified by the second and fourth authors. The elaboration of the study has followed the Declaration of Helsinki as a statement of ethical principles for medical research involving human subjects. 

## 3. Results

Two main themes and four subthemes emerged from the analysis of the data. All of them help us to describe and understand adolescents’ experience of sexuality and its influencing factors. 

### 3.1. Sexuality and Sex Education in Late Adolescence

The participants of our study recognised the importance of sexual aspects in the development of an individual’s life and the influence they have on many areas of our lives. This theme describes the role that sexual behaviour plays in our lives, the importance of sex education, and the determining factors for a healthy sexual development. 

#### 3.1.1. The Impact of Sexual Behaviour on the Development of the Lives of Adolescents

According to the opinion of our participants, sex is a fundamental and necessary element for good physical and mental development in their lives because of its direct effect on the individual. Sex is a source of well-being, releasing tensions, and providing pleasure or satisfaction. In addition, sexual relationships are used to satisfy basic physical needs or simply as a form of entertainment. 


*“Sex provides pleasure, satisfaction and helps to reach levels of ecstasy, as well as releasing tension and feeling better about yourself”*
P1.


*“These types of (sexual) behaviour are usually essential throughout most human beings’ lives, whether for entertainment or simply out of necessity”*
P7.

Many participants report that sexual behaviour affects the physiological aspect of human life and is essential in social relationships as they are a way to connect with and get to know other people. Many adolescents use sex to forge intimacy, which in turn generates trust in another person. Similarly, sex allows us to know and explore ourselves and becomes a way of defining ourselves. Our participants even report that sex is even necessary to forge and maintain a relationship. 


*“I consider sex to be a significant part of our relationships with other people, and I think they also speak to how we are and what has happened to us”*
P5.


*“Sexual behaviour are very important as they allow the couple to become more and more trusting”*
P8.

#### 3.1.2. The Absence of Basic Sexual Education

Sex education allows us to enjoy free sexuality and rationally avoid the risks associated with it, such as sexually transmitted diseases or unwanted pregnancies. Adolescents reported that sex education was insufficient, even irrelevant. They described this education as null and void on the part of parents, and reduced to basic talks in the academic context, without depth and contextualisation. This lack of sex education can lead to irrational thoughts and false beliefs, focusing on irrational fears of pregnancy or contracting sexually transmitted diseases. 


*“My sexual education has been lousy; I have not received enough information either at the educational level or at the family level. The information I have obtained about sex education has been based on some talks given at the educational centre, but with little foundation, and the rest of the information I have sought myself through different means”*
P11.

In contrast, only one of the participants seemed to have benefited from the sexual education received in the family context due to the mutual trust and naturalness of talking about sexuality with his parents.


*“I have had a sexual education which in my opinion is very complete since I have obtained a lot of information, mostly from my parents”*
P8.

The adolescents in our study obtained information about sexuality in a self-taught way, researching on their own via the internet or other sources such as books or via their own experience or that of other friends.


*“Sexual education has been obtained through experience. No close person has given me information about sex”*
P13.


*“My sexual education has been good but mainly because I have done my own research, I have read anatomy books and also looked on the internet about the questions that I have in my head”*
P4.

### 3.2. Societal Influences on Sexual Life 

The second theme identified was the role that society plays in the sexual lives of individuals as it is present during the development of the life cycle. The influence of parental figures, the peer group, or the pressures of the environment itself ultimately create social norms that condition sexuality. 

#### 3.2.1. The Influence of Parents and the Peer Circle on Sexual Aspects

The family context is a determining element in personal development, including the sexual sphere, with a greater influence in the early years of life. Parents, therefore, have a great influence on sexual education as they are the first figures to provide information and guidance. However, participants described that the role of parents as sex educators could be counterproductive, especially if they transmitted a negative and restrictive vision of sexual life. This repressive context of sex creates a climate of mistrust and discomfort between parents and children. The behaviour of our participants’ parents tended to hide everything related to sex, treating it as something bad and making irrational judgments about their sons and daughters and of their relationships. In addition, they try to prohibit attitudes and behaviours in sexual relations and impose the need to have a regular partner as a prerequisite for sexual relations. As a result, and with greater influence in adolescence, the family environment creates tensions in the development of sexual life, a period of great physical change that is crucial to the formation of personality. These attitudes thus transmit misinformation, fears, and insecurities to their children. 


*“I never tell my parents anything regarding my sexual activity as they have a very different view of it than I do. They consider that in order to have sex you must necessarily be in a couple, so they would see it as wrong for me to have sex or engage in sexual practices outside of the couple. Anyway, I consider that they have never given me enough confidence to talk to them about such issues, and that it would lead me to feel judged by them if I told them about my sex life or my relationships”*
P9.


*“They have never talked openly about the subject; in fact, when in some TV series there are images of sex, they usually change the channel or look the other way. The fact they do this conveys to me that sex is something bad, that you have to hide it, when in fact it is not”*
P11.

However, many participants described the mother figure as the main source of support and trust in sexual matters. Greater contact with the maternal figure and a more sensitive and less restrictive approach to sexual matters are essential for the good development of adolescents’ sexual lives. 


*“Certain sexual things I usually tell them, but to my mother, because my father is usually more distant. I feel freer telling her, and she also gives me confidence to talk about all this”*
P7.


*“I think it (sex) is not something to hide, you have to talk about it with total normality. I have enough confidence with my mum to be able to talk about it and being understood. I get the best advice from my mum”*
P12.

The peer group plays a determining role in sexual development throughout an individual’s life. Most of the adolescents interviewed had a diverse intimate circle. Friendships acted as support figures and as an outlet to which participants could confide their sexual experiences and insecurities. This is because friends are in a similar position: they tend to be non-judgmental and seek the best form of advice on sexuality.


*“I have people I talk to about my sexual activity, mostly friends in my inner circle. I trust them because they give me the confidence to talk about any topic and for them sex is not a taboo subject, and they understand perfectly what I can contact them about it, without judging me or giving me an opinion about the way I should develop my sexual life, but they look for the best way to advise me”*
P2.

Participants with a more reserved personality had a very small circle of friends and hardly any people they trusted, so they treated sexual issues as something intimate, and if they were in a couple, they considered it disrespectful to the partner to talk about sexual issues outside the couple.


*“I consider sex to be an intimate part of my life, and I don’t see the point in sharing it with anyone”*
P10.


*“I am not a person who usually tells my intimacies to people out of respect for my partner”*
P5.

Participants in our study reported that, in certain circumstances, peer groups that ridiculed, disrespected, or belittled a sexual experience or attitude could have negative consequences by creating insecurities, feelings of shame, emotional discomfort, and even feelings of inferiority. On these occasions, the participants described the peer group as “false friendships” because no one should judge or ridicule an opinion, behaviour, attitude, or feeling and become repressive about sexual matters.


*“I get angry because I feel that when they laugh at my sexual behaviour they are belittling it, and I think no one should belittle other people’s way of living sexuality”*
P10.

#### 3.2.2. Repression of Sex in Society and the Influence of Social Pressure

The social context determines the sexual life of individuals from birth to the end of the life cycle. Even today, most social contexts share a conservative morality and treat sex as a taboo subject. Our adolescents mentioned that, as a result, behaviour was judged, and sexual behaviour was imposed, leading them to act according to norms in order to avoid being discriminated against or judged. Extremist attitudes, such as treating sex as a sin, a generalized fear of sexual behaviour, or exclusivity in male–female relationships without accepting other types of relationships, have a negative impact on the free development of sexuality. According to the opinion of our participants, these conceptions and attitudes end up forcing them to act in ways that are sometimes the opposite of what they really want.


*“In the end I have to hide what I want to do, if I want to do something, and for different reasons I can’t do them, this will affect me more psychologically than physically because I would always be thinking about that thing. This would lead to me not being myself if I ever felt sexually repressed”*
P4.

Adolescents highlighted the importance of virginity as a very influential element in our society. Virginity is an overvalued social construct in which there is great pressure to lose it with the right person and at the right time. According to the participants, virginity is simply a myth that should be treated as a subjective and less important moment to be lost in any situation where the person feels comfortable. 


*“I think in this sense it is very old-fashioned that you have to lose your virginity with your partner, or that you have to wait for the “right” person”*
P2.


*“I don’t think there is a set age for losing virginity. A lot of people panic about reaching a certain age and not having lost it. I think it’s something you should lose when you think it’s time and not get carried away by society”*
P9.

However, today’s society does not have a negative attitude towards sexuality. The adolescents in our study told us that, little by little, the taboo on sexuality is gradually being removed as its centrality and naturalness in human beings is being reinforced. The participants emphasized the need to adopt positions where the freedom to behave as they wish is respected, respecting each person’s right to choose and the freedom to behave as we wish.


*“In my environment I do not consider sex a taboo subject because I can talk about it with total freedom. And it is true that in general sex can no longer be considered a taboo subject because there are various media such as social networks where we can talk about it with total freedom”*
P6.


*“I think it used to be more of a taboo subject than it is now. Little by little this topic (sex) is becoming normalized, and people are less embarrassed to talk about it either with friends or with parents”*
P4.

## 4. Discussion

The aim of this study was to explore and describe the perspectives of late adolescents on the main influences on the development of their sexual lives. The participants considered how sexual life plays a crucial role in their personal and social lives via its influence on physical, cognitive, psychosocial, and affective-motivational development. These findings seem to be consistent with other studies, which conclude that the most important factors in an individual’s sexual development and maturation are the family, peer group, sexual education, and social contexts [16,17].

Along the same lines, Seiler-Ramadas et al. (2019) stated that their participants perceived that the sex education received from their parents and educational centres was limited and reductionist [18]. However, both the educational and family environments are fundamental for the formation and development of basic sexual competencies in adolescents [19,20]. Moreover, family passivity in sexual education is associated with poor communication and distance between the family and the school environment in sexual matters [21]. In contrast, a comprehensive sexual education at school and in the home provides information on a wide range of psychosocial and health issues and also allows for greater mobility, leading to changes in safe and healthy sexual behaviour [22,23]. However, only a minority of educational settings currently provide this type of comprehensive sexual education [24,25]. A comprehensive sexual education is considered a human right and aims to provide accurate and realistic information and life skills in a non-judgmental way to help adolescents make informed choices [26]. In the absence of information, adolescents turn to other sources of information such as the sexual experiences of friends and self-education via the internet and books [27,28].

Although the influence of parents is a determining factor in the sexual development of adolescents as they are the first sources of information and guidance, in many cases, parents as sex educators become detrimental to the full development of adolescents’ sexuality because they transmit a negative and restrictive view of sexuality [19]. In these cases, an atmosphere of mistrust and discomfort is created between parents and children, where everything related to sexuality is hidden [29]. However, it should be noted that, as our results show, when the mother fulfils her role as a sexual educator, there is generally a break in the restrictive view of sexuality, and she becomes a source of support, a result consistent with the study by Yamanaka & Kawata (2020) [30]. 

Faced with the negative sexual situation of their parents, adolescents turn to their peers as support and venting figures, sharing experiences, information, thoughts, and beliefs, although not all peers can be trusted with their sexual experiences [31]. In our results, adolescents create intimate circles with their most trusted peers. In addition, the wider circle of friends loses influence, although they remain trusted and intimate figures [32,33]. 

Finally, our results describe the influence of the social context that, due to the dominance of conservative morality rooted in religion, generates social rejection of the sexual and imposes social norms by which sexual behaviour is judged and considered abnormal if it deviates from these social norms. Social taboos regarding sexuality are very strict and intolerant, mainly due to religious and social mandates [34,35,36]. Virginity has great importance in our society, and adolescents feel pressured to lose it with the ideal people and at the right moment, when it is just a myth, a social construct that has been overvalued by society. 

This study has some limitations. The results of this study are influenced by cultural and social patterns in western societies. In order to compare the results, similar studies should be conducted in other societies. All participants had a heterosexual orientation, so the results of our study may have differed in late adolescents with a different sexual orientation. Furthermore, this study only collected experiences and opinions of late adolescents, and further studies should include the opinions of parents and teachers.

## 5. Conclusions

Sexuality is a fundamental part of our lives and influences the construction of our personal development. Adolescence is a time of change when we are more susceptible to external influences that ultimately determine adolescents’ sexuality. The most influential environments in late adolescent sexuality are the social, family, school, and peer contexts. However, due to the influence of a predominantly conservative morality rooted in religion, the environment provides a negative view of sexuality and represses sexual behaviour, limiting sexual education and focusing it on fear. Adolescents are, therefore, forced to turn to their peers and to self-education for information and to resolve their sexual doubts. It is necessary to change attitudes towards sexuality, to normalize sexual diversity, and to provide the necessary skills to manage sexual life and ensure healthy sexual development without fear or repression. 

## Figures and Tables

**Table 1 healthcare-11-03032-t001:** Sociodemographic characteristics of the participants.

Participant	Gender	Age	Sexual Orientation	Type of Relationship
1	Male	19	Heterosexual	Open
2	Male	20	Heterosexual	Open
3	Female	20	Heterosexual	Closed
4	Male	21	Heterosexual	Closed
5	Female	17	Heterosexual	Without relationship
6	Male	17	Heterosexual	Open
7	Male	19	Heterosexual	Closed
8	Female	18	Heterosexual	Open
9	Female	18	Heterosexual	Open
10	Female	21	Heterosexual	Closed
11	Female	17	Heterosexual	Closed
12	Female	19	Heterosexual	Closed
13	Male	18	Heterosexual	Closed

**Table 2 healthcare-11-03032-t002:** Units of meaning, themes, and subthemes.

Units of Meaning	Subthemes	Themes
Well-being, bodily needs, self-knowledge, knowledge of another person, way of defining ourselves, confidence	1.1. The impact of sexual behaviours on the development of the lives of adolescents.	1. Sexuality and sex education in late adolescence
Insufficient information, self-taught education, experience, internet information	1.2. The absence of basic sexual education	
Importance of the partner, influence of education, repressive contexts, support figures.	2.1. The influence of parents and the peer circle on sexual aspects.	2. Societal influences on sexual life.
Naturalization of sexuality, taboo environments, overvalued virginity, dominant behaviours and thoughts, conservative society.	2.2. Repression of sex in society and the influence of social pressure.	

## Data Availability

Data are contained within the article.

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
