# Peer review of "Factors Influencing Sexuality in Late Adolescence: A Qualitative Study on Heterosexual Adolescents’ Perspectives"

_healthcare, 2023, doi:10.3390/healthcare11233032_

Round 1

Reviewer 1 Report

Comments and Suggestions for Authors

Suggestions for textual changes: 

Title

Lines 1-2 Factors influencing sexuality in late adolescence: A qualitative study on heterosexual adolescents’ perspectives. 

Inclusion of heterosexual as all 13 participants recruited for the study identified as heterosexual.

Abstract: 

Line 13  The results of our study confirm offer further support of the importance of sexual aspects in our the lives of late adolescents who identify as heterosexual 

Introduction

Line 23 Change -Sexuality, which accompanies us throughout our life cycle, is an integral part of adolescent health, with great importance in for their physical, emotional and social 

Materials and Methods: 

Line 73-74 Seven of them had an open relationship, five a closed relation ship, and one did not have any kind of relationship.

Query: Re: Type of Relationship: What do these categories of open/closed refer to? Are these sexually intimate relationships? If so, then need to state that.

Results

Line 142 ... for a good physical and mental development in our their lives due to its direct

Line 161 --3.1.2. The absence of basis basic sexuality education 

Lines 203-204 -  Consequently, and with greater influence in adolescence, the family environment generates troubles for the development of sexual life, and this is a period of great physical changes, which is critical in the formation of personality. 

This sentence needs to be rephrased for clarity e.g. generates tensions

Line 257 ...leading us them to act according to norms to avoid being...

Line 262 ...forcing us them to act in a way that is sometimes the opposite of what we they really want.

Line 281 ...the freedom to behave as we they want.

General Queries

How is the date of the University Ethics Approval 03/03/2023 (Line 361) after the stated date for data collection - January and February 2022 (Line 83)

Given that the study explores the role of society as a potential factor influencing adolescent sexual development - it would be helpful to have an overview of the social context for the adolescent participants in this study. Not all social contexts across the globe are similar and an account of this study's social context would support further interpretation of the findings and their application to similar or other social contexts.

Comments on the Quality of English Language

Needs minor textual revision as noted.

Author Response

Title

Lines 1-2 Factors influencing sexuality in late adolescence: A qualitative study on heterosexual adolescents’ perspectives. 

Inclusion of heterosexual as all 13 participants recruited for the study identified as heterosexual.

Thank you for your comments. We have included that all 13 participants recruited for the study identified as heterosexual. 

Abstract: 

Line 13  The results of our study confirm offer further support of the importance of sexual aspects in our the lives of late adolescents who identify as heterosexual 

We are very grateful for your comments. We have changed the line 13 in the abstract.

Introduction

Line 23 Change -Sexuality, which accompanies us throughout our life cycle, is an integral part of adolescent health, with great importance in for their physical, emotional and social 

Thank you for your comment. The sentence has been changed. 

Materials and Methods: 

Line 73-74 Seven of them had an open relationship, five a closed relation ship, and one did not have any kind of relationship.

Query: Re: Type of Relationship: What do these categories of open/closed refer to? Are these sexually intimate relationships? If so, then need to state that.

Thank you for your comment. Type of relationship refer to the adolescents had intimate relationships with only a partner or more.

Results

Line 142 ... for a good physical and mental development in our their lives due to its direct

Thank you for your comment. The sentence has been changed.

Line 161 --3.1.2. The absence of basis basic sexuality education 

Thank you for your comment. The sentence has been changed. 

Lines 203-204 -  Consequently, and with greater influence in adolescence, the family environment generates troubles for the development of sexual life, and this is a period of great physical changes, which is critical in the formation of personality. 

This sentence needs to be rephrased for clarity e.g. generates tensions

Thank you for your comment. The sentence has been changed. 

Line 257 ...leading us them to act according to norms to avoid being...

Thank you for your comment. The sentence has been changed.

Line 262 ...forcing us them to act in a way that is sometimes the opposite of what we they really want.

Thank you for your comment. The sentence has been changed. 

Line 281 ...the freedom to behave as we they want.

Thank you for your comment. The sentence has been changed. 

General Queries

How is the date of the University Ethics Approval 03/03/2023 (Line 361) after the stated date for data collection - January and February 2022 (Line 83)

We are very grateful for your comments and opinions. Sorry for the mistake, it has been corrected.

Given that the study explores the role of society as a potential factor influencing adolescent sexual development - it would be helpful to have an overview of the social context for the adolescent participants in this study. Not all social contexts across the globe are similar and an account of this study's social context would support further interpretation of the findings and their application to similar or other social contexts.

Thank you for your comment. The social context is clarify in the introduction section.

Reviewer 2 Report

Comments and Suggestions for Authors

It seems logical that since this is a qualitative study, both an ethics committee (EC), albeit a social one, and an informed consent in word/paper format, which should be submitted together with the approval of the EC, should be taken into account. If this was not done, at least an EC must approve that the conduct of this research followed the ethical standards underpinning research per the Declaration of Helsinki.

With respect to the data obtained from informants, it is recommended that the transcribed data be reviewed by each of the informants, which is not detailed.

Likewise, in the discussion, it is requested that the bibliographical references be increased for greater consistency in the research.

The limitations and explanations expressed in this section are gratefully acknowledged.

I encourage the authors to revise the article as its improvement will provide an excellent scientific publication.

Author Response

It seems logical that since this is a qualitative study, both an ethics committee (EC), albeit a social one, and an informed consent in word/paper format, which should be submitted together with the approval of the EC, should be taken into account. If this was not done, at least an EC must approve that the conduct of this research followed the ethical standards underpinning research per the Declaration of Helsinki.

Thank you for your comment. We have included that the study has followed the etical principles of Declaration of Helsinki. 

With respect to the data obtained from informants, it is recommended that the transcribed data be reviewed by each of the informants, which is not detailed.

We are very grateful for your comments and opinions. We have included that transcribed data have been reviewed by the informants.

Likewise, in the discussion, it is requested that the bibliographical references be increased for greater consistency in the research.

Thank you for your comments. We have included more bibliographical references, but the evidence about the topic is limited. 

The limitations and explanations expressed in this section are gratefully acknowledged.

I encourage the authors to revise the article as its improvement will provide an excellent scientific publication.

Reviewer 3 Report

Comments and Suggestions for Authors

An interesting topic and area of study but the justification of the study must be stronger to show the new contribution the study was adding to the literature on adolescent sexuality.

Comments on the Quality of English Language

Well written with minimal editing.

Author Response

Thank you for your comments and opinions. The manuscript has been improved. 

Round 2

Reviewer 2 Report

Comments and Suggestions for Authors

I thank the authors for their efforts in making the modifications.

The article has gained clarity and improved its final result.

Comments on the Quality of English Language

I thank the authors for their efforts in making the modifications.

The article has gained clarity and improved its final result.

Author Response

Thank you for your comments and suggestions. The manuscript has received editing of English language. 

Reviewer 3 Report

Comments and Suggestions for Authors

Abstract needs to be written for clarity and understanding.

Line 31 to 42: Contextual factors. I suggest interpersonal relationships.

Line 85-88: Sentence too long it lost its meaning. I suggest it be rewritten for clarity. Why the use of interviews and not focus groups, etc.?

Line 280: “However, not all is negative in today’s society.” What do you mean?

Line 344: a word “of” is missing.

Comments on the Quality of English Language

There is improved clarity except for the abstract.

Author Response

Abstract needs to be written for clarity and understanding.

Thank you for your comments. Abstract has been rewritten.

Line 31 to 42: Contextual factors. I suggest interpersonal relationships.

We are very grateful for your comments and suggestions. The word “contextual factors” has been replaced y “interpersonal relationships”.

Line 85-88: Sentence too long it lost its meaning. I suggest it be rewritten for clarity.

Why the use of interviews and not focus groups, etc.?

Thank you for your suggestions. The sentence has been rewritten. We use interviews and not focus groups because the participants preferred to speak alone about this topic.

Line 280: “However, not all is negative in today’s society.” What do you mean?

Thank you for your comment. The sentence has been rewritten and clarify.

Line 344: a word “of” is missing.

Thank you for your comment. The missing word “of” has been incorporated.
